# Structural Analysis of Brain Hub Region Volume and Cortical Thickness in Patients with Mild Cognitive Impairment and Dementia

**DOI:** 10.3390/medicina56100497

**Published:** 2020-09-24

**Authors:** Nauris Zdanovskis, Ardis Platkājis, Andrejs Kostiks, Guntis Karelis

**Affiliations:** 1Department of Radiology, Riga East University Hospital, Hipokrāta iela 2, LV-1038 Riga, Latvia; ardis.platkajis@rsu.lv; 2Department of Radiology, Riga Stradins University, Dzirciema iela 16, LV-1007 Riga, Latvia; 3Department of Neurosurgery and Neurology, Riga East University Hospital, Hipokrāta iela 2, LV-1038 Riga, Latvia; andrejs.kostiks@gmail.com (A.K.); guntis.karelis@gmail.com (G.K.)

**Keywords:** brain networks, brain hubs, mild cognitive impairment, dementia, neuroimaging, cortical thickness, white matter volume

## Abstract

*Background and Objectives*: A complex network of axonal pathways interlinks the human brain cortex. Brain networks are not distributed evenly, and brain regions making more connections with other parts are defined as brain hubs. Our objective was to analyze brain hub region volume and cortical thickness and determine the association with cognitive assessment scores in patients with mild cognitive impairment (MCI) and dementia. *Materials and Methods*: In this cross-sectional study, we included 11 patients (5 mild cognitive impairment; 6 dementia). All patients underwent neurological examination, and Montreal Cognitive Assessment (MoCA) test scores were recorded. Scans with a 3T MRI scanner were done, and cortical thickness and volumetric data were acquired using Freesurfer 7.1.0 software. *Results*: By analyzing differences between the MCI and dementia groups, MCI patients had higher hippocampal volumes (*p* < 0.05) and left entorhinal cortex thickness (*p* < 0.05). There was a significant positive correlation between MoCA test scores and left hippocampus volume (r = 0.767, *p* < 0.01), right hippocampus volume (r = 0.785, *p* < 0.01), right precuneus cortical thickness (r = 0.648, *p* < 0.05), left entorhinal cortex thickness (r = 0.767, *p* < 0.01), and right entorhinal cortex thickness (r = 0.612, *p* < 0.05). *Conclusions*: In our study, hippocampal volume and entorhinal cortex showed significant differences in the MCI and dementia patient groups. Additionally, we found a statistically significant positive correlation between MoCA scores, hippocampal volume, entorhinal cortex thickness, and right precuneus. Although other brain hub regions did not show statistically significant differences, there should be additional research to evaluate the brain hub region association with MCI and dementia.

## 1. Introduction

The human brain cortex is interlinked by a complex network of axonal pathways that range from smaller local circuits and broader long-range fiber pathways [1,2]. There are several structural features of cortical networks that can be utilized as a quantitative variable through graph theory, i.e., nodal degree, strength, eccentricity, path length, clustering coefficient, transitivity, centrality, etc. [3,4,5] Brain networks are not distributed evenly. Brain hubs are the parts of the brain that are making many connections with other parts of the brain. [6,7] Brain network hub functionality is essential for neuronal communication and integration [6,8]. These hubs are located in the bilateral putamen, thalamus, superior parietal, superior frontal, precuneus, hippocampus, insula, right pallidum, and left lingual gyrus [7].

Several studies found a brain network hub connectivity disruption association with mild cognitive impairment and Alzheimer’s disease [9,10,11].

We analyzed brain network hub volume and cortical thickness and determined whether there are distinct differences in these regions for patients with mild cognitive impairment (MCI) and dementia. Additionally, we analyzed entorhinal cortex cortical thickness that is considered as a reliable Alzheimer’s disease brain biomarker [12,13].

## 2. Materials and Methods

Participants were admitted to the neurological outpatient clinic due to suspected cognitive impairment. Patients underwent neurological examination, and the Montreal Cognitive Assessment (MoCA) test was performed, and scores were recorded. Patients were divided into two groups—mild cognitive impairment and dementia. The average and median age and MoCA scores are shown in Table 1.

Exclusion criteria for patients were other clinically significant neurological diseases, and drug or alcohol abuse. Patients did not have any other significant MR abnormalities (e.g., tumors, malformations, large vessel stroke).

MRI was performed on a single-site scanner to avoid scanner differences. All patients were scanned on a 3T scanner. We used the GE MP-RAGE sequence protocol with 1 mm^3^ resolution and 1 mm slice thickness with the appropriate gray–white matter contrast that was evaluated for every patient. All scans were converted from DICOM format to NIFTI format to perform further analysis.

Cortical reconstruction and volumetric segmentation were performed by using Freesurfer 7.1.0 image analysis software. It is documented and freely available for download online (http://surfer.nmr.mgh.harvard.edu/). The technical details of these procedures are described in prior publications [14,15,16,17,18,19,20,21,22,23,24,25,26,27,28,29].

We used the Desikan–Killiany–Tourville (DKT) labeling protocol to extract cortical thickness results [18].

Data were analyzed with statistical analysis software JASP Version 0.13. Descriptive statistics for volumetric data and cortical thickness data were determined in the MCI and dementia groups. The Mann–Whitney U test was used to analyze brain hub volume and cortical thickness differences in the MCI and dementia groups. Spearman’s correlation was calculated, and statistical significance was determined in both groups by correlating MoCA scores and hub volume and cortical thickness data.

Ethics committee (3 October 2019) and institutional review board (21 October 2019) approvals were obtained (ethical approval number: AP-144/19). Written informed consent for participation in a study and use of anonymous data was obtained for every patient.

## 3. Results

### 3.1. Mean Values of Volumes and Cortical Thickness

Mean values of volumes and cortical thickness with standard deviation and standard error were calculated and are shown in Table 2.

### 3.2. The Mann–Whitney U Test

By analyzing differences between MCI and dementia groups, statistically significant results were found in both hemisphere hippocampal volumes and left entorhinal cortex thickness (*p* < 0.05). Other regions did not show statistically significant differences between the MCI and dementia groups. Multiple comparison correction was not performed when reporting *p* values. Thus, results serve as exploratory data that must be validated with a larger cohort and further multiple comparison correction. The results are presented in Table 3.

The median value for **left hippocampal volume** (Table 4) in the MCI group was 3742.1 mm^3^, and in the dementia group 2938.0 mm^3^, and the distribution in these two groups differed significantly (Mann–Whitney U = 27, *p* = 0.03).

The median value for **right hippocampal volume** (Table 4) in the MCI group was 4004.0 mm^3^, and in the dementia group 2995.4 mm^3^, and the distribution in these two groups differed significantly (Mann–Whitney U = 28, *p* = 0.02).

The median value for **left entorhinal cortex thickness** (Table 4) in the MCI group was 2.896 mm, and in the dementia group 2.226 mm, and the distribution in these two groups differed significantly (Mann–Whitney U = 28, *p* = 0.02).

### 3.3. Spearman’s Correlations

The Spearman’s correlations (Table 5) were conducted to determine whether there are associations of the MoCA score and volume or cortical thickness in hub regions. The two-tailed test of significance indicated that there was a significant positive correlation between MoCA score and left hippocampus volume (r = 0.767, *p* < 0.01), right hippocampus volume (r = 0.785, *p* < 0.01), right precuneus cortical thickness (r = 0.648, *p* < 0.05), left entorhinal cortex thickness (r = 0.767, *p* < 0.01), and right entorhinal cortex thickness (r = 0.612, *p* < 0.05).

## 4. Discussion

In this study, we compared brain hub region structural data and entorhinal cortex thickness in patients with MCI and dementia. We found that out of all brain hub regions, only hippocampal volume had statistically significant differences in both groups. Additionally, we analyzed entorhinal cortex thickness in the MCI and dementia groups and found statistically significant differences in the left entorhinal cortex.

So, to analyze the brain hub region association with MCI and dementia, we performed MoCA score, volume, and cortical thickness associations analysis. We found a significant positive correlation in both hemispheres’ hippocampal volume, right precuneus cortical thickness, and both hemispheres’ entorhinal cortex thickness with MoCA scores.

Many studies have found smaller hippocampal volumes in patients with dementia than MCI or healthy controls [30,31].

Precuneus atrophy is associated with Alzheimer’s dementia [32] and dementia in Parkinson’s disease [33].

Further, the entorhinal cortex is proposed as a biomarker for early detection of Alzheimer’s disease. [10,11].

Regarding putamen structural changes, there are mixed results. One study found that putamen had strongly reduced volumes in Alzheimer’s disease [34], but also in another study, putamen structural values did not diverge from normal cognition patients across the entire lifespan [35].

Other brain hub regions have been analyzed separately and associated with cognitive decline, MCI, or dementia [35,36].

By analyzing our data, we did not find significant differences in the MCI and dementia groups apart from the structures mentioned above. This study was exploratory rather than confirmatory and performed on a small cohort. We did not perform multiple comparison correction when reporting *p* values. We are planning to validate our results and perform multiple comparison corrections with a larger cohort.

## 5. Conclusions

In our study, hippocampal volume and entorhinal cortex showed significant differences in the MCI and dementia patient groups. Additionally, we found a statistically significant positive correlation between MoCA scores, hippocampal volume, entorhinal cortex thickness, and right precuneus.

Although other brain hub regions did not show statistically significant differences, there should be additional research to evaluate the brain hub region association with MCI and dementia.

## Figures and Tables

**Table 1 medicina-56-00497-t001:** Demographic data and Montreal Cognitive Assessment (MoCA) scores in study patients.

	Age		MoCA	
	MCI	Dementia	MCI	Dementia
N	5	6	5	6
Mean	62.0	69.5	25.4	11.7
Median	62.0	71.0	25.0	12.0
Std. Deviation	10.6	2.7	2.5	4.9
Minimum	48.0	66.0	23.0	4.0
Maximum	77.0	72.0	28.0	18.0

**Table 2 medicina-56-00497-t002:** Mean values with standard deviation and standard error comparing patients with mild cognitive impairment (MCI) and dementia.

	Group	N	Mean	SD	SE
Left Hippocampus Volume, mm^3^	MCI	5	4048.620	453.702	202.902
	Dementia	6	2882.800	652.183	266.253
Right Hippocampus Volume, mm^3^	MCI	5	4209.600	699.158	312.673
	Dementia	6	3059.567	541.566	221.093
Left Pallidum Volume, mm^3^	MCI	5	1796.220	262.356	117.329
	Dementia	6	1605.667	241.104	98.430
Right Pallidum Volume, mm^3^	MCI	5	1754.340	235.098	105.139
	Dementia	6	1593.517	190.760	77.877
Left Putamen Volume, mm^3^	MCI	5	4479.360	642.134	287.171
	Dementia	6	3652.117	548.654	223.987
Right Putamen Volume, mm^3^	MCI	5	4497.800	616.991	275.927
	Dementia	6	4031.617	305.439	124.695
Left Thalamus Volume, mm^3^	MCI	5	6802.060	1187.686	531.150
	Dementia	6	5936.450	940.564	383.984
Right Thalamus Volume, mm^3^	MCI	5	6756.820	1097.194	490.680
	Dementia	6	5970.650	681.772	278.332
Left Superior Parietal Gyrus Cortical Thickness, mm	MCI	5	2.153	0.096	0.043
	Dementia	6	2.160	0.129	0.053
Right Superior Parietal Gyrus Cortical Thickness, mm	MCI	5	2.102	0.110	0.049
	Dementia	6	2.095	0.123	0.050
Left Superior Frontal Gyrus Cortical Thickness, mm	MCI	5	2.438	0.098	0.044
	Dementia	6	2.452	0.142	0.058
Right Superior Frontal Gyrus Cortical Thickness, mm	MCI	5	2.410	0.111	0.050
	Dementia	6	2.462	0.131	0.053
Left Precuneus Cortical Thickness, mm	MCI	5	2.280	0.069	0.031
	Dementia	6	2.245	0.214	0.088
Right Precuneus Cortical Thickness, mm	MCI	5	2.280	0.115	0.051
	Dementia	6	2.179	0.144	0.059
Left Insula Cortical Thickness, mm	MCI	5	2.960	0.234	0.105
	Dementia	6	2.851	0.202	0.083
Right Insula Cortical Thickness, mm	MCI	5	2.969	0.206	0.092
	Dementia	6	2.824	0.128	0.052
Left Lingual Gyrus Cortical Thickness, mm	MCI	5	1.962	0.144	0.064
	Dementia	6	2.000	0.060	0.025
Right Lingual Gyrus Cortical Thickness, mm	MCI	5	1.957	0.180	0.081
	Dementia	6	1.975	0.077	0.031
Left Entorhinal Cortex Cortical Thickness, mm	MCI	5	2.896	0.304	0.136
	Dementia	6	2.226	0.349	0.143
Right Entorhinal Cortex Cortical Thickness, mm	MCI	5	2.986	0.482	0.215
	Dementia	6	2.515	0.511	0.209

**Table 3 medicina-56-00497-t003:** The Mann–Whitney U test by comparing the MCI and dementia patient groups.

	W	*p*
Left Hippocampus Volume, mm^3^	27.000	0.030 *
Right Hippocampus Volume, mm^3^	28.000	0.017 *
Left Pallidum Volume, mm^3^	22.000	0.247
Right Pallidum Volume, mm^3^	22.000	0.247
Left Putamen Volume, mm^3^	25.000	0.082
Right Putamen Volume, mm^3^	22.000	0.247
Left Thalamus Volume, mm^3^	21.000	0.329
Right Thalamus Volume, mm^3^	21.000	0.329
Left Superior Parietal Gyrus Cortical Thickness, mm	14.000	0.931
Right Superior Parietal Gyrus Cortical Thickness, mm	16.500	0.855
Left Superior Frontal Gyrus Cortical Thickness, mm	14.000	0.931
Right Superior Frontal Gyrus Cortical Thickness, mm	13.000	0.792
Left Precuneus Cortical Thickness, mm	16.000	0.931
Right Precuneus Cortical Thickness, mm	22.000	0.247
Left Insula Cortical Thickness, mm	15.000	1.000
Right Insula Cortical Thickness, mm	22.000	0.247
Left Lingual Gyrus Cortical Thickness, mm	13.000	0.784
Right Lingual Gyrus Cortical Thickness, mm	13.000	0.792
Left Entorhinal Cortex Cortical Thickness, mm	28.000	0.017 *
Right Entorhinal Cortex Cortical Thickness, mm	24.000	0.126

* *p* < 0.05.

**Table 4 medicina-56-00497-t004:** Median values in the left hippocampus, right hippocampus, and left entorhinal cortex.

	Left Hippocampus Volume, mm^3^		Right Hippocampus Volume, mm^3^		Left Entorhinal Cortex Cortical Thickness, mm	
	MCI	Dementia	MCI	Dementia	MCI	Dementia
N	5	6	5	6	5	6
Mean	4048.620	2882.800	4209.600	3059.567	2.896	2.226
Median	3742.100	2938.200	4004.000	2995.400	2.896	2.260
Std. Deviation	453.702	652.183	699.158	541.566	0.304	0.349
Minimum	3708.800	1882.700	3370.600	2395.800	2.428	1.773
Maximum	4683.100	3764.900	5166.600	3966.300	3.214	2.711

**Table 5 medicina-56-00497-t005:** Spearman’s correlation coefficients and *p* values by correlating MoCA score with volume and cortical thickness.

Variable		MoCA
1. MoCA	Spearman’s rho	-
	*p*-value	-
2. Left Hippocampus Volume, mm^3^	Spearman’s rho	0.767 **
	*p*-value	0.006
3. Right Hippocampus Volume, mm^3^	Spearman’s rho	0.785 **
	*p*-value	0.004
4. Left Pallidum Volume, mm^3^	Spearman’s rho	0.443
	*p*-value	0.172
5. Right Pallidum Volume, mm^3^	Spearman’s rho	0.584
	*p*-value	0.059
6. Left Putamen Volume, mm^3^	Spearman’s rho	0.470
	*p*-value	0.144
7. Right Putamen Volume, mm^3^	Spearman’s rho	0.589
	*p*-value	0.057
8. Left Thalamus Volume, mm^3^	Spearman’s rho	0.374
	*p*-value	0.257
9. Right Thalamus Volume, mm^3^	Spearman’s rho	0.333
	*p*-value	0.316
10. Left Superior Parietal Gyrus Cortical Thickness, mm	Spearman’s rho	0.169
	*p*-value	0.619
11. Right Superior Parietal Gyrus Cortical Thickness, mm	Spearman’s rho	0.304
	*p*-value	0.363
12. Left Superior Frontal Gyrus Cortical Thickness, mm	Spearman’s rho	−0.260
	*p*-value	0.440
13. Right Superior Frontal Gyrus Cortical Thickness, mm	Spearman’s rho	−0.283
	*p*-value	0.399
14. Left Precuneus Cortical Thickness, mm	Spearman’s rho	0.123
	*p*-value	0.718
15. Right Precuneus Cortical Thickness, mm	Spearman’s rho	0.648 *
	*p*-value	0.031
16. Left Insula Cortical Thickness, mm	Spearman’s rho	−0.055
	*p*-value	0.873
17. Right Insula Cortical Thickness, mm	Spearman’s rho	0.192
	*p*-value	0.572
18. Left Lingual Gyrus Cortical Thickness, mm	Spearman’s rho	−0.092
	*p*-value	0.789
19. Right Lingual Gyrus Cortical Thickness, mm	Spearman’s rho	−0.005
	*p*-value	0.989
20. Left Entorhinal Cortex Cortical Thickness, mm	Spearman’s rho	0.767 **
	*p*-value	0.006
21. Right Entorhinal Cortex Cortical Thickness, mm	Spearman’s rho	0.612 *
	*p*-value	0.045

* *p* < 0.05, ** *p* <0.01.

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
