# Peer review of "Structural Analysis of Brain Hub Region Volume and Cortical Thickness in Patients with Mild Cognitive Impairment and Dementia"

_medicina, 2020, doi:10.3390/medicina56100497_

Round 1
Reviewer 1 Report
1 of 2:
Regarding multiple comparisons:
"The multiple comparison correction was not performed as it can provide false negatives (there could be changes between groups, but those would be flagged as statistically non-insignificant). On such a small cohort multiple comparison correction would definitely have a huge impact. Thus, we are planning to do multiple comparison corrections when we will have a larger cohort"
The converse is that avoiding a correction can lead to false positives. This is acceptable but must be stated in the text of the manuscript, not just our correspondence. See for example "Science Forum: Ten common statistical mistakes to watch out for when writing or reviewing a manuscript": https://elifesciences.org/articles/48175?_ga=2.87814296.1615571170.1583746607-392198663.1550050583
In particular in the "Failing to correct for multiple comparisons" section: "Exploratory testing can be absolutely appropriate, but should be acknowledged".
The authors need to state in the text of manuscript that the p values reported have not been corrected for multiple comparisons. For example, when you discuss limitations, include: "Because this study was exploratory rather than confirmatory, and performed on a small cohort, we did not perform any form of multiple comparison correction when reporting p values. We are planning to do multiple comparison corrections when we will have a larger cohort."
2 of 2:
Regarding "clinical significance", do not use the phrase "clinically significant" to describe results with p < 0.05. A p value has nothing to do with clinical significance. Instead use the phrase "statistically significant". See for example "Common pitfalls in statistical analysis: Clinical versus statistical significance": https://www.ncbi.nlm.nih.gov/pmc/articles/PMC4504060/. Clinical significance involves "the extent of change, whether the change makes a real difference to subject lives, how long the effects last, consumer acceptability, cost-effectiveness, and ease of implementation".
The use of "clinically significant" is appropriate on line 56: "Exclusion criteria for patients were other clinically significant neurological diseases"
The use is not appropriate on lines 87 and 89: "clinically significant results were
found in both hemisphere hippocampal volumes and left entorhinal cortex thickness (p<0.05). Other regions did not show clinically significant differences"
It is not appropriate on line 116: "clinically significant results were found in both hemisphere hippocampal volumes and left entorhinal cortex thickness (p<0.05). Other regions did not show clinically significant differences"
Please replace these 3 occurrences with "statistically significant"
Author Response
Dear reviewer,
Thank You for Your time and suggestions.
I added sentences where it is mentioned that we did not perform multiple comparisons when reporting p values (line 90-92) and in the discussion added that this study serves as an exploratory and multiple comparisons were not performed (line 138-140).
I changed "clinically significant" to "statistically significant" where it is appropriate (line 87, 89, 118).
Also, thank you for the articles that you shared - good material for further articles.
Have a nice day!
Best wishes,
Nauris Zdanovskis
Reviewer 2 Report
The authors accomplished all my requests.
Author Response
Dear reviewer,
Thank You for Your time and effort and useful suggestions!
Have a nice day!
Best wishes,
Nauris Zdanovskis
This manuscript is a resubmission of an earlier submission. The following is a list of the peer review reports and author responses from that submission.
Round 1
Reviewer 1 Report
Summary:
This paper discusses volume and thickness differences in brain hub regions (and entorhinal cortex) between patients with MCI (n=5) and dementia (n=6), as well as correlations with MoCA test scores. Measurements were made from MRI with Freesurfer. Several brain regions were reported to show significant group differences or correlations with MoCA.
General Comments:
As many studies have investigated these relationships, in groups with much larger sample sizes, the paper's novelty is limited (see for example the Alzheimer's Disease Neuroimaging Initiative: http://adni.loni.usc.edu/news-publications/publications/). If this small population is particularly relevant or novel, it should be stated clearly.
Because multiple comparisons during hypothesis testing are a particularly strong problem in neuroimaging studies, focusing on hub regions rather than the whole brain is an interesting approach. However, there is no description of how multiple comparison correction was performed in this study despite reporting p values for 20 different regions. If multiple comparison correction was performed, it should be stated clearly. Several of my evaluation metrics would be improved in this case ("Are the methods adequately described", "Are the conclusions supported by the results", "Scientific soundness"). If multiple comparison correction was not performed, many of the conclusions are likely not significant and the results would have to be re evaluated.
Half the references are to FreeSurfer's technical details, and some of these are for components that were not used (such as longitudinal analysis). The relevance of these references should be reexamined. The authors should include references to more closely related work, and speak to their specific contributions to the field, and whether their results affirm or contradict the existing body of literature.
Overall the paper is structured clearly and concisely. While there are several grammatical errors, I do not have any line by line specific comments.
Author Response
Dear reviewer,
Thank You for Your time and review!
This study is still ongoing and in the future, we plan to expand our patient database (including controls).
The multiple comparison correction was not performed as it can provide false negatives (there could be changes between groups, but those would be flagged as statistically non-insignificant). On such a small cohort multiple comparison correction would definitely have a huge impact. Thus, we are planning to do multiple comparison corrections when we will have a larger cohort.
As far as I understand, it is a mandatory condition to use FreeSurfer - the technical references should be included (https://surfer.nmr.mgh.harvard.edu/fswiki/FreeSurferMethodsCitation). Also, for the readers who are not familiar with FreeSurfer technical details could benefit from all the references.
We removed the longitudinal analysis reference by Reuter et al.
In the discussion section, we added additional comparisons with the literature data.
Best regards,
Nauris Zdanovskis
Reviewer 2 Report
Authors produced an interesting research about a potential biomarker for Mild Cognitive Impairment and Dementia patients based on structural features from brain magnetic resonance imaging. The work proposed a method to discriminate mild cognitive impairment (MCI) from dementia patients, a common problem in the current neuropathological differential diagnosis. It took advantage of complex brain network theory insights, namely the notion of network hubs.
Main contributions: The proposed method estimated the volume and the thickness of cortical regions identified as brain network hubs (i.e. regions highly connected with other ones). The methods seemed effective although validated on a small population of patients.
Findings: They found that MCI had greater hippocampal volume (both hemispheres) and left entorhinal cortical thickness. In addition, the MoCa score was positively correlated with other relevant regions such as the precuneus, the hippocampus and the entorhinal cortex.
Strengths: The method, whether will be validated on a larger cohort of patients on future developments, could be effectively inserted in the clinical practice to discriminate mild cognitive impairment (MCI) from dementia patients.The major limitation of the study is the very limited number of patients (N=11) and the used study population that did not contained a control group. Therefore, I would strongly suggest to soften the conclusions.
I would like to add one more suggestion for authors. The should refer to a more general definition of "brain networks" in the introduction because it was often associated with "cortical" while they proposed a methodology based also on subcortical regions (i.e. hippocampus).
Author Response
Dear reviewer,
Thank You for Your time and provided comments.
We are recruiting new patients as our study goes further and we will validate results on a larger cohort of patients.
As per your suggestions, we softened the conclusions.
The last suggestion was noted and our definition has been changed to "brain networks" from "brain cortical networks".
Best regards,
Nauris Zdanovskis